# The Use of Static Posturography Cut-Off Scores to Identify the Risk of Falling in Older Adults

**DOI:** 10.3390/ijerph19116480

**Published:** 2022-05-26

**Authors:** Agnieszka Wiśniowska-Szurlej, Agnieszka Ćwirlej-Sozańska, Anna Wilmowska-Pietruszyńska, Bernard Sozański

**Affiliations:** 1Institute of Health Sciences, Medical College, Rzeszow University, Warzywna 1A Street, 35-310 Rzeszów, Poland; asozanska@ur.edu.pl; 2Homes of Medical Care Rehabilitation Center Donum Corde, Budy Głogowskie 835B Street, 36-060 Głogów Małopolski, Poland; 3Faculty of Medicine, Lazarski University, Świeradowska 43 Street, 02-662 Warsaw, Poland; anna.wilmowska@autograf.pl; 4Institute of Medicine, Medical College, Rzeszow University, Warzywna 1A Street, 35-310 Rzeszów, Poland; bsozanski@ur.edu.pl

**Keywords:** aged, diagnostic techniques, postural balance

## Abstract

Background: Falling is the most common accident that occurs in daily living and the second leading cause of unintentional injury death worldwide. The complexity of the risk factors associated with falling makes older people at risk of falling difficult to identify. The aim of the study was to identify the cut-off scores of standing posturography measures that can be used to predict the risk of falling in older adults. Methods: This observational study involved 267 elderly people aged 65 to 85 years (73.99 SD 7.51) living in south-eastern Poland. The subjects were divided into two groups: a group with a high risk of falling and a group with a low risk of falling, based on their timed up-and-go test. Postural stability was assessed during eyes-open and eyes-closed trials using the two-plate stability platform CQ Stab 2P. Results: The best accuracy, sensitivity, and specificity were observed for the sway path, anterior-posterior sway path, and medial-lateral sway path with open and closed eyes. The clinical cut-off score to predict the risk of falling was 350.63 for the sway path with open eyes, 272.64 for the anterior-posterior sway path, and 159.63 for the medial-lateral sway path. The clinical cut-off score for sway path with closed eyes was 436.11. Conclusions: Static posturography screenings in clinical practice may also be useful for detecting typical balance changes in older adults.

## 1. Introduction

Falling is the most common accident that occurs in daily living and the second leading cause of unintentional injury death in the world [1]. Approximately 28–35% of people aged 65 years and older fall every year, and this result increases to 32–42% for people over 70 years of age [2]. In older people, falls can lead to severe injuries such as hip fracture or head trauma [3]. The length of hospitalization in older people is nine days longer for fall-related reasons than for other reasons. Moreover, older adults hospitalized due to fall-related injuries are more often moved into long-term care facilities [4]. These transfers result in a loss of independence in everyday activities for the individual and increased healthcare system costs.

Although falls can occur at any age, their frequency increases with age. Falls that are experienced by older people result from involutional changes (physical, perceptual, and cognitive), together with an inappropriate environment regarding safety [5]. There are many validated tools to predict the risk of falling in the elderly. According to the systematic review of Seong-Hi Park, the Berg Balance Scale (BBS) and Timed Up and Go (TUG) test were generally used for the community-dwelling older people as fall risk assessment tools [6]. To identify older people at increased risk of future falls, studies have most commonly used a history of falls and the assessment of impairments in gait, mobility, and balance. Although the systematic assessment of the risk of falls in the elderly is recommended in geriatric medicine, the tools currently used do not allow for a reliable and objective measurement result [7].

Research indicates that postural instability is best studied with objective posturographic measurements, the advantages of which, over regular clinical evaluation, include reducing the variability of test scores and avoiding subjective scoring systems [8]. Posturography provides information about specific balance control mechanisms and thus constitutes a clinically useful tool to identify fall risk. Posturographic tests use force plates to measure the movement of the foot center of pressure (CoP) while the subject stands in a relaxed position on the platform with his eyes fixed at one point [9]. The difficulty of taking the measurement can be enlarged by eliminating the visual control. The eyes closed increased the postural sway in people of all ages compared with eyes open; however, the differences are most distinguishable in older adults. Maintaining the correct balance of body posture contributes to the prevention of falls and affects the longer functional independence of older people [10].

The studies with the use of posturography have shown that it is an objective and quantitative measure of balance deficits, and they have confirmed its important clinical application in the assessment of the risk of falling in older people [11,12]. Some sway characteristics of stance, especially in the mediolateral direction, significantly differ between non-fallers and fallers and may therefore be good indicators of individuals at increased risk of future falls. To identify the differences between falling and non-falling older people, studies have assessed anterior-posterior (AP) plantar centre of pressure (CoP) movements and medial-lateral (ML) sway amplitude with eyes open and closed [13,14]. Additionally, Pizzigalli et al. indicated that path length CoP, CoP velocity, and sway in the AP and ML directions are the variables that distinguish older adult fallers from non-fallers [15]. A larger sway when the eyes are closed is characteristic of people who have fallen repeatedly [16]. Studies have also shown differences in automated postural responses between older men and women [17]. According to Howcroft et al., a posturography examination of the standing position in older adults is a useful screening tool for older people who are at risk of falling [18].

The authors emphasize that objective measures of postural sway independently predict incident falls in older women and men [8]. Despite the widespread use of stabilometric measurements, there is currently no consensus about what values of features should be used for diagnostics [19]. Therefore, additional studies are needed to determine the postural stability values important for predicting falls in a clinical setting. Measures of postural sway predict falls in the elderly. It is important to identify people most at risk of falling in order to implement the necessary precautions and prevent (or at least slow down) age-related causes of falls by means of training and rehabilitation [20].

To the best of our knowledge, this is the first study in Poland presenting eyes open and eyes closed standing posturography with older adults to determine cut-off scores for risk of falling. The aims of the study were to assess postural stability with the eyes open and closed in older women and men and to establish the cut-off scores that can be used to identify people with an increased risk of falling.

## 2. Materials and Methods

### 2.1. Design

This observational study was carried out among a randomly selected population of older people. The study was approved by the Bioethics Committee of the University of Rzeszow (Resolution No. 4/3/2017). In accordance with the Declaration of Helsinki, the participants were provided with information about the aim and the course of the study and expressed their written and informed consent to participate.

This study was performed and reported according to the Strengthening the Reporting of Observational Studies in Epidemiology Criteria (STROBE) [21].

### 2.2. Participants and Setting

This observational study involved 267 elderly people aged 65 to 85 years (73.99 SD 7.51) living in south-eastern Poland. Community-living older people who volunteered to participate in the study were recruited by announcements on the regional radio, at local churches, at universities of the third age, at senior clubs, and at day-care nursing homes.

The participants were included in the study based on the following inclusion criteria: an age of 65 years or older, a normal cognitive status corresponding to an abbreviated mental test score (AMTS) > 6 points, the level of physical performance required by the participants to take a standing position on the stabilometric platform, and the provision of informed consent for participation in the study. The exclusion criteria were as follows: vestibular and neurological disorders, dizziness, the consumption of drugs that significantly affect balance, injuries of the lower limbs that occurred within the last 6 months, paresis or deformities in the upper limbs, and severe systemic diseases.

The subjects were divided into two groups.

A group with a high risk of falling (HRF), which included 136 people with a high risk of falling, based on their timed up-and-go test (TUG) results (≥13.5 s), and a group without risk of falling (WRF), which included 131 people with a timed up-and-go test result <13.5.

A faster TUG time indicates better functional performance, and a score of ≥13.5 s was used as the cut-off point to identify those at increased risk of falls in the community setting [22].

The tests were performed in the laboratory of gerontoprophylaxis in the Center for Innovative Research in Medical and Natural Sciences (Poland).

### 2.3. Outcome Measurements

Basic sociodemographic data (sex, age, body mass, height) were collected during the study. Body mass index (BMI) was calculated as weight in kg divided by height in meters squared and classified according to World Health Organization categories [23].

The American Geriatric Society and the British Geriatric Society recommend the TUG as a routine screening test for falls in older people [24]. Therefore, the TUG test was used to assess the risk of falling in the study group. During the TUG test, the patient is timed while they rise from an armchair, walk at a comfortable and safe pace for a distance of 3 m, turn and walk back to the chair and sit again. The test was carried out three times. The trial with the shortest time was selected for the assessment.

Postural stability was assessed by the use of the two-plate stability platform CQ Stab 2P (CQ Elektronik System, Czernica, Poland). The device allowed the vertical CoP positions of the forces affecting each foot to be recorded simultaneously. Each of the platform plates had three force sensors that determined the displacement of the center of pressure on the support plane. During the measurements, the values describing the static balance were recorded. The platform plates were placed parallel, 2 m from the wall of the room; there was a mark on the wall for the participants to fix their eyesight during the test with open eyes. Before each trial, the device was calibrated. The study protocol featured two 30-s successive tests. The first test was conducted to assess the body’s stability with the eyes wide open (EO), and the second test was conducted while the subject had his or her eyes closed (EC). The proper measurement was preceded by a 30-s “training” session in which the participant maintained balance, and then the test readings were recorded. The subjects were instructed to remove their shoes and stand freely on the platform plates (each foot resting on a separate panel of the platform) with their arms relaxed along the trunk. The feet were placed parallel to each other, keeping a steady distance between the feet (8 cm between 1 metatarsal bone and the mid-platform edge line; the lateral ankles were placed along the vertical line to the line dividing the platform into two halves). The higher the value of the parameters recorded by the platform, the larger the CoP displacement was in the support plane [25].

The following parameters were used in the analysis:Sway path—total path length measured on the XY axes in mm;Anterior-posterior sway path—statokinesiogram path length measured on the *Y*-axis direction in mm;Medial-lateral sway path—statokinesiogram path length measured on the *X*-axis in mm;Mean Amplitude—mean CoP displacement (radius) in mm;Anterior-posterior mean amplitude—mean CoP displacement from point 0 in the *Y*-axis direction in mm;Medial-lateral mean amplitude—mean CoP displacement from point 0 in the *X*-axis direction in mm;Maxima anteroposterior sway CoP—maximal CoP displacement from point 0 in the *Y*-axis direction in mm;Maximal lateral sway CoP—maximal CoP displacement from point 0 in the *X*-axis direction in mm.

### 2.4. Statistical Analysis

The data were analyzed using statistical software R, version 4.0.2, with the ROC i DiscriMiner package (R Foundation for Statistical Computing, Vienna, Austria). For the continuous variables, the mean and standard deviation were reported, and percentages were used for the categorical variables. The descriptive data for the current study population were stratified by sex and risk of falling. The subject characteristics were compared using the Mann–Whitney U test for the continuous variables and the chi-square test for the categorical variables. *p*-values < 0.05 were considered significant.

For variables that were significantly different between HFR and WFR, cut-off scores for faller classification were determined using the clinical cut-off score method [26], receiver operating characteristic (ROC) curves [27], and discriminant functions [28]. Three different techniques were used for analysis to identify the strongest cut-off score.

## 3. Results

The study included 267 people, including 142 women and 125 men. The mean age was 73.99 years (SD 7.51), while the BMI was 27.49 (5.29). A total of 131 people without risk of falling and 136 people with a high risk of falling were examined. The study groups did not differ from each other in terms of basic anthropometric and sociodemographic parameters. Table 1 presents the characteristics of the study groups.

The results revealed statistically significant differences in all the examined parameters of postural stability, both with EO and with EC between the groups differing by the risk of falling. People at high risk of falling had significantly higher sway values, and this result was evident among the women as well. The largest difference was observed in the sway path (EC 606.32 vs. EO 261.76). However, among the men, no differences were found between HRF and WRF in the mean amplitude, anterior-posterior mean amplitude, medial-lateral mean amplitude, maximal anteroposterior sway CoP, and maximal lateral sway CoP with EC (Table 2).

The best accuracy, sensitivity, and specificity were achieved for the sway path, anterior-posterior sway path, and medial-lateral sway path with EO and EC. The clinical cut-off score for sway path with EO was 350.63, which means that people with values higher than this score have a high risk of falling (80.52% accuracy, 84.56% sensitivity, and 76.34% specificity); that for the anterior-posterior sway path with EO was 272.64, and that for the medial-lateral sway path was 159.63. The ROC cut-off score was 368.50 for the sway path (79.40% accuracy, 80.15% sensitivity, and 78.63% specificity), 272.50 for the anterior-posterior sway path with EO, and 149.50 for the medial-lateral sway path. The clinical cut-off score for the sway path with EC was 436.11, and the ROC cut-off score was 359.50. The discriminant function cut-off score for a high risk of falling was 3.03 (Table 3).

The females showed the best accuracy, sensitivity, and specificity for the sway path, anterior-posterior sway path, and medial-lateral sway path with EO and EC. The clinical cut-off score for the sway path with EO was 349.73 (85.21% accuracy, 86.49% sensitivity, and 83.82% specificity), for the anterior-posterior sway path with EO was 264.39, and for the medial-lateral sway path was 174.33. The ROC cut-off score for the sway path was 364.00 (82.39% accuracy, 81.08% sensitivity, and 83.82% specificity), for the anterior-posterior sway path with EO was 272.50, and for the medial-lateral sway path was 151.00. The clinical cut-off score for the sway path with EC was 396.14, and the ROC cut-off score was 349.00. The discriminant function cut-off score for a high risk for falling was 4.00 (Table 4).

The males showed the best accuracy, sensitivity, and specificity for the sway path, anterior-posterior sway path, and medial-lateral sway path with EO. The clinical cut-off score for the sway path with EO was 358.06 (76.00% accuracy, 80.65% sensitivity, and 71.43% specificity), for the anterior-posterior sway path with EO was 286.78, and for the medial-lateral sway path was 151.77. The ROC cut-off score for the sway path was 365.00 (76.00% accuracy, 80.65% sensitivity, and 71.43% specificity), for the anterior-posterior sway path with EO was 268.50, and for the medial-lateral sway path was 148.00. The discriminant function cut-off score for a high risk for falling was 2.80 (Table 5).

## 4. Discussion

Center of pressure trajectory parameters need to be identified to classify and predict people at risk of falling and to conduct objective and long-term monitoring of the most vulnerable elderly individuals. Recent advancements in the collection and storage of data enable posturographic data to be used to determine the magnitude of improvement in an individual’s ability to maintain balance in the process of rehabilitation and his or her status.

The results of this study show that posturographic data can distinguish people with a high risk of falling from people without a risk of falling. The largest differences were related to the parameters of the sway path, anterior-posterior sway path, and medial-lateral sway path. Similar results were obtained by Merlo et al. The authors showed that the mean positions of the CoP in the AP and ML directions were associated with fall history in older adults [29]. A systematic review by Quijoux et al. showed that prospective studies previously demonstrated that differences in the mean velocities in the AP and ML directions are significantly different between fallers and non-fallers. The ML features were less consistent and discriminative than the AP features in the present analysis [30].

Among the examined women, all analyzed posturographic parameters, both with eyes open and eyes closed, differed statistically significantly between HRF and WRF. In the men, a relationship was demonstrated between posturographic parameters with eyes open and the risk of falling. However, with eyes closed, no differences were found in the following parameters: mean amplitude, anterior-posterior mean amplitude, medial-lateral mean amplitude, maximal anteroposterior sway CoP, and maximal lateral sway CoP. A prospective study by Cella et al. showed that stabilometric assessments can more accurately identify older people at high risk of falls, thus enabling more personalized fall prevention interventions [7].

Objective measures of postural sway allow for predicting the risk of falling, although there are indeed no stability limits that can identify those who are most vulnerable in this regard. The results of the prospective observational study from the Healthy Aging Initiative cohort indicated that the parameters of postural sway would allow for an accurate fall risk prediction, whereas the outcomes of further work should set normative values in order to identify the most vulnerable people [31].

The research showed the highest sensitivity, specificity, and accuracy for the sway path. The area under the curve was the highest for a sway path of 0.89. The clinical cut-off score for sway path with EO was 350.63, and the area under the ROC was 368.50. According to Johansson et al., there was a 75% increase in fall risk in older adults with CoP sway lengths ≥400 mm in the test with open eyes [8], while Prosperini et al. indicated that the risk of falling may grow by 8% to each 10-mm increase in the CoP pathway with eyes open [11].

In the group of women, the clinical cut-off score for sway path EO was 349.73 (ROC—364.00), while for the men, the clinical cut-off score for sway path with EO was 358.06 (ROC—365.00). This result means that people with the abovementioned results have a high risk of falling. Our study provides a basis for identifying, monitoring, and managing interventions for elderly people at risk of falling.

Additionally, high values were shown for the anterior-posterior sway path. The clinical cut-off score for the sway path with EO was 272.64, and the area under the ROC curve was 272.50. In the group of women, the clinical cut-off score for the sway path with EO was 264.39 (ROC—272.50), while for the men, the clinical cut-off score for the sway path with EO was 286.78 (ROC—268.50). As we observed higher values of sensitivity, specificity, and accuracy for the clinical cut-off scores, the clinical cut-off score method demonstrated a greater predictive value than the ROC method.

This study has limitations. Due to the observational nature of the study, the idea that balance impairment precedes falls cannot be confirmed. The validity of the estimated cut-off points should be verified in a longitudinal study. Additionally, the limitation may be a convenient sample of relatively healthy older adults. Another limitation of the study is the lack of monitoring of actual falls, as well as of existing chronic diseases and medications.

The strength of this study is that, to the best of our knowledge, it is the first research assessing postural stability with the eyes open and closed in older women and men and establishing the cut-off scores that can be used to identify people with an increased risk of falling. Although several studies had previously agreed that older fallers exhibit higher CoP displacements than older non-fallers, the ranges distinguishing people with an increased risk of falling still remained undefined. Therefore, this study helps fill an existing gap for rehabilitation experts by providing cut-off scores of postural balance measures for screening community-dwelling older adults for a high risk of falling. Identifying the most vulnerable people to a fall risk will help to implement proper training at an early stage. According to the systematic review of Sherrington et al., performing exercises reduces the frequency of falls by 23% [32]. Various forms of training, including multi-component exercise training [33], Tai Chi [34], or functional training with Total Resistance Exercises (TRX), lead to positive changes in the static and dynamic balance of older adults [35].

## 5. Conclusions

Stabilometric measurements can be used to identify people at high risk of falling. In both women and men, statistically significant differences in the CoP trajectory between people with HRF and WRF were found. Cut-off scores can be used to identify older people who may fall in the future.

## Figures and Tables

**Table 1 ijerph-19-06480-t001:** Participant characteristics.

	Total	High Risk of Falling	Without Risk of Falling
**Total**	267	136	131
**Female**	142	74	68
**Male**	125	62	63
**Age**	73.99 (7.51)	74.83 (8.53)	73.12 (7.31)
**Body Mass Index**	27.49 (5.29)	27.32 (6.01)	27.64 (4.43)
**Timed Up and Go**	18.01 (10.40)	25.24 (10.01)	10.51 (2.40)

**Table 2 ijerph-19-06480-t002:** Mean, standard deviation, and *p*-value for groups differing by the risk of falling under eyes open and closed conditions.

Measures	Total			Female			Male		
	High Risk of Falling Me (Q1–Q3)	Without Risk of Falling Me (Q1–Q3)	*p*-Value	High Risk of Falling Me (Q1–Q3)	Without Risk of Falling Me (Q1–Q3)	*p*-Value	High Risk of Falling Me (Q1–Q3)	Without Risk of Falling Me (Q1–Q3)	*p*-Value
**Sway Path O**	504.50 (380.50–729.50)	265.00 (199.00–345.00)	<0.001	551.50 (377.00–774.00)	242.50 (187.50–313.00)	<0.001	474.50 (389.00–688.00)	279.00 (228.00–381.00)	<0.001
**Anterior-Posterior Sway Path O**	394.00 (305.50–593.00)	197 (140–272)	<0.001	401.00 (306.00–605.00)	174.00 (131.00–243.00)	<0.001	378.50 (295.00–581.00)	224.00 (161.00–312.00)	<0.001
**Medial-Lateral Sway Path O**	210.00 (157.00–317.00)	126.00 (102.00–155.00)	<0.001	210.00 (160.00–334.00)	122.00 (100.00–157.00)	<0.001	207.50 (152.00–313.00)	126.00 (106.00–155.00)	<0.001
**Mean Amplitude O**	5.35 (4.10–7.60)	3.90 (2.60–5.20)	<0.001	5.55 (4.30–7.70)	3.65 (2.15–4.55)	<0.001	5.05 (3.80–7.50)	4.30 (2.90–5.60)	0.005
**Anterior-Posterior Mean Amplitude**	4.00 (3.00–5.20)	2.80 (1.80–3.70)	<0.001	4.00 (3.20–4.90)	2.75 (1.50–3.45)	<0.001	4.10 (2.80–5.70)	2.80 (1.90–4.30)	<0.001
**Medial-Lateral Mean Amplitude**	2.70 (1.65–4.30)	1.60 (0.90–2.50)	<0.001	2.80 (1.90–4.40)	1.25 (0.80–2.15)	<0.001	2.55 (1.60–4.10)	2.1 (1.3–2.8)	0.023
**Maxima anteroposterior sway CoP O**	16.20 (11.50–22.05)	9.70 (6.80–13.10)	<0.001	16.70 (12.10–21.90)	9.55 (6.50–11.95)	<0.001	15.45 (10.40–24.00)	10.40 (7.30–16.30)	<0.001
**Maximal lateral sway CoP O**	10.65 (5.80–18.75)	6.00 (3.30–8.60)	<0.001	11.25 (6.10–19.60)	5.00 (2.60–7.70)	<0.001	9.15 (5.60–18.30)	7.00 (4.90–9.00)	0.011
**Sway Path C**	547.00 (404.50–728.00)	306.00 (235.00–419.00)	<0.001	541.00 (400.00–717.00)	278.50 (205.00–379.00)	<0.001	550.00 (419.00–734.00)	342.00 (264.00–512.00)	<0.001
**Anterior-Posterior Sway Path C**	436.00 (297.50–644.00)	246.00 (172.00–346.00)	<0.001	433.00 305.00–638.00)	218.00 (159.00–279.00)	<0.001	439.00 (290.00–669.00)	288.00 (211.00–451.00)	<0.001
**Medial-Lateral Sway Path C**	204.50 (158.00–296.00)	128.00 (106.00–164.00)	<0.001	205.50 (161.00–296.00)	120.50 (97.00–153.00)	<0.001	200.50 (156.00–304.00)	135 (107–167)	<0.001
**Mean Amplitude C**	4.75 (3.40–6.65)	3.60 (2.50–4.70)	<0.001	4.40 (3.30–6.50)	2.90 (1.90–4.00)	<0.001	5.00 (3.40–70)	4.10 (2.90–6.30)	0.060
**Anterior-Posterior Mean Amplitude C**	3.75 (2.65–5.20)	2.80 (2.10–3.90)	<0.001	3.70 (2.70–4.90)	2.40 (1.65–3.30)	0.425	4.15 (2.40–5.50)	3.40 (2.50–5.20)	0.425
**Medial-Lateral Mean Amplitude C**	2.00 (1.40–3.10)	1.60 (0.70–2.80)	0.001	1.95 (1.40–3.00)	0.95 (0.60–2.05)	<0.001	2.00 (1.50–3.20)	2.40 (1.30–3.20)	0.923
**Maxima anteroposterior sway CoP C**	15.55 (10.55–22.85)	10.40 (6.60–15.80)	<0.001	15.80 (9.70–22.70)	8.85 (5.70–12.60)	<0.001	15.55 (10.60–24.20)	12.50 (9.30–19.40)	0.056
**Maximal lateral sway CoP C**	7.60 (4.90–12.55)	5.50 (3.10–8.70)	<0.001	7.15 (4.80–12.60)	4.00 (2.65–6.70)	<0.001	7.85 (5.40–12.00)	7.50 (4.80–9.40)	0.130

Note: O—open eyes; C—closed eyes; CoP—plantar center of pressure.

**Table 3 ijerph-19-06480-t003:** Clinical, ROC, and discriminant function cut-off scores for a high risk for falling (total).

Method	Measure	Cut-Off Score	Accuracy (%)	Sensitivity (%)	Specificity (%)
Clinical	Sway Path O	350.63	80.52	84.56	76.34
	Anterior-Posterior Sway Path O	272.64	77.90	80.15	75.57
	Medial-Lateral Sway Path O	159.63	74.91	73.53	76.34
	Mean Amplitude O	4.55	65.54	64.71	66.41
	Anterior-Posterior Mean Amplitude	3.42	67.79	66.18	69.47
	Medial-Lateral Mean Amplitude	2.53	65.17	53.68	77.10
	Maxima anteroposterior sway CoP O	13.25	70.79	65.44	76.34
	Maximal lateral sway CoP O	9.39	67.79	56.62	79.39
	Sway Path C	436.11	73.41	69.85	77.10
	Anterior-Posterior Sway Path C	364.36	70.79	63.97	77.86
	Medial-Lateral Sway Path C	170.38	71.16	63.97	78.63
	Mean Amplitude C	4.18	61.42	58.82	64.12
	Anterior-Posterior Mean Amplitude C	3.68	62.17	52.94	71.76
	Medial-Lateral Mean Amplitude C	2.39	54.31	43.38	65.65
	Maxima anteroposterior sway CoP C	14.86	62.17	53.68	70.99
	Maximal lateral sway CoP C	8.03	57.68	48.53	67.18
ROC	Sway Path O (AUC = 0.897)	368.50	79.40	80.15	78.63
	Anterior-Posterior Sway Path O (AUC = 0.885)	272.50	77.90	80.15	75.57
	Medial-Lateral Sway Path O (AUC = 0.853)	149.50	76.40	80.88	71.76
	Mean Amplitude O (AUC = 0.726)	3.75	64.42	81.62	46.56
	Anterior-Posterior Mean Amplitude (AUC = 0.735)	2.75	64.42	80.15	48.09
	Medial-Lateral Mean Amplitude (AUC = 0.711)	1.55	64.42	80.88	47.33
	Maxima anteroposterior sway CoP O (AUC = 0.766)	10.15	65.92	80.15	51.15
	Maximal lateral sway CoP O (AUC = 0.723)	5.45	61.05	80.15	41.22
	Sway Path C (AUC = 0.793)	359.50	71.91	80.15	63.36
	Anterior-Posterior Sway Path C (AUC = 0.763)	265.00	68.54	80.15	56.49
	Medial-Lateral Sway Path C (AUC = 0.787)	139.50	70.79	80.15	61.07
	Mean Amplitude C (AUC = 0.677)	2.95	61.80	82.35	40.46
	Anterior-Posterior Mean Amplitude C (AUC = 0.637)	2.35	58.43	81.62	34.35
	Medial-Lateral Mean Amplitude C (AUC = 0.613)	1.25	60.67	80.88	39.69
	Maxima anteroposterior sway CoP C (AUC = 0.675)	9.45	61.42	80.15	41.98
	Maximal lateral sway CoP C (AUC = 0.668)	4.55	61.42	80.15	41.98
Discriminant Function	Formula	3.03	78.65	66.91	90.84

Note: O—open eyes; C—closed eyes; CoP—plantar center of pressure; AUC—Area Under Curve. Discriminant Function Formula: 0.0088 · Sway Path O + 0.0001 · Anterior-Posterior Sway Path O—0.0093 · Medial-Lateral Sway Path O—0.3495 · Mean Amplitude O + 0.1707 · Anterior-Posterior Mean Amplitude + 0.2743 · Medial-Lateral Mean Amplitude + 0.0172 · Maxima anteroposterior sway CoP O + 0.032 · Maximal lateral sway CoP O + 0.0009 · Sway Path C—0.0028 · Anterior-Posterior Sway Path C + 0.0061 · Medial-Lateral Sway Path C + 0.3188 · Mean Amplitude C—0.3165 · Anterior-Posterior Mean Amplitude C—0.2871 · Medial-Lateral Mean Amplitude C + 0.0417 · Maxima anteroposterior sway CoP C + 0.008 · Maximal lateral sway CoP C.

**Table 4 ijerph-19-06480-t004:** Clinical, ROC, and discriminant function cut-off scores for a high risk for falling (females).

Method	Measure	Cut-Off Score	Accuracy (%)	Sensitivity (%)	Specificity (%)
Clinical	Sway Path O	349.73	85.21	86.49	83.82
	Anterior-Posterior Sway Path O	264.39	81.69	82.43	80.88
	Medial-Lateral Sway Path O	174.33	76.06	68.92	83.82
	Mean Amplitude O	4.19	71.83	78.38	64.71
	Anterior-Posterior Mean Amplitude	3.13	69.01	75.68	61.76
	Medial-Lateral Mean Amplitude	2.49	67.61	56.76	79.41
	Maxima anteroposterior sway CoP O	12.56	73.94	70.27	77.94
	Maximal lateral sway CoP O	8.95	73.24	64.86	82.35
	Sway Path C	396.14	76.06	75.68	76.47
	Anterior-Posterior Sway Path C	320.26	77.46	74.32	80.88
	Medial-Lateral Sway Path C	173.14	71.13	62.16	80.88
	Mean Amplitude C	3.56	64.79	68.92	60.29
	Anterior-Posterior Mean Amplitude C	3.25	66.90	59.46	75.00
	Medial-Lateral Mean Amplitude C	1.93	59.86	50.00	70.59
	Maxima anteroposterior sway CoP C	13.10	68.31	60.81	76.47
	Maximal lateral sway CoP C	7.37	62.68	50.00	76.47
ROC	Sway Path O (AUC = 0.928)	364.00	82.39	81.08	83.82
	Anterior-Posterior Sway Path O (AUC = 0.927)	272.50	81.69	81.08	82.35
	Medial-Lateral Sway Path O (AUC = 0.862)	151.00	76.76	81.08	72.06
	Mean Amplitude O (AUC = 0.798)	3.95	70.42	82.43	57.35
	Anterior-Posterior Mean Amplitude (AUC = 0.777)	2.85	68.31	83.78	51.47
	Medial-Lateral Mean Amplitude (AUC = 0.785)	1.65	74.65	82.43	66.18
	Maxima anteroposterior sway CoP O (AUC = 0.817)	11.20	73.24	81.08	64.71
	Maximal lateral sway CoP O (AUC = 0.788)	5.30	67.61	81.08	52.94
	Sway Path C (AUC = 0.841)	349.00	77.46	81.08	73.53
	Anterior-Posterior Sway Path C (AUC = 0.837)	268.00	76.06	81.08	70.59
	Medial-Lateral Sway Path C (AUC = 0.81)	139.50	73.94	81.08	66.18
	Mean Amplitude C (AUC = 0.761)	2.95	69.72	85.14	52.94
	Anterior-Posterior Mean Amplitude C (AUC = 0.737)	2.35	64.79	82.43	45.59
	Medial-Lateral Mean Amplitude C (AUC = 0.728)	1.05	70.42	86.49	52.94
	Maxima anteroposterior sway CoP C (AUC = 0.754)	9.00	66.90	81.08	51.47
	Maximal lateral sway CoP C (AUC = 0.756)	4.35	69.01	81.08	55.88
Discriminant Function	Formula	4.00	83.80	75.68	92.65

Note: O—open eyes; C—closed eyes; CoP—plantar center of pressure; AUC—Area Under Curve. Discriminant Function Formula:—0.001 · Sway Path O + 0.0101 · Anterior-Posterior Sway Path O—0.0001 · Medial-Lateral Sway Path O—0.9414 · Mean Amplitude O + 0.6434 · Anterior-Posterior Mean Amplitude + 0.8114 · Medial-Lateral Mean Amplitude + 0.0155 · Maxima anteroposterior sway CoP O + 0.027 · Maximal lateral sway CoP O + 0.0149 · Sway Path C—0.0131 · Anterior-Posterior Sway Path C—0.0118 · Medial-Lateral Sway Path C + 0.9368 · Mean Amplitude C—0.6653 · Anterior-Posterior Mean Amplitude C—0.4394 · Medial-Lateral Mean Amplitude C + 0.0283 · Maxima anteroposterior sway CoP C—0.0042 · Maximal lateral sway CoP C.

**Table 5 ijerph-19-06480-t005:** Clinical, ROC, and discriminant function cut-off scores for a high risk for falling (males).

Method	Measure	Cut-Off Score	Accuracy (%)	Sensitivity (%)	Specificity (%)
Clinical	Sway Path O	358.06	76.00	80.65	71.43
	Anterior-Posterior Sway Path O	286.78	73.60	75.81	71.43
	Medial-Lateral Sway Path O	151.77	75.20	77.42	73.02
	Mean Amplitude O	5.13	56.80	48.39	65.08
	Anterior-Posterior Mean Amplitude	3.74	62.40	54.84	69.84
	Medial-Lateral Mean Amplitude	2.58	60.00	50.00	69.84
	Maxima anteroposterior sway CoP O	13.97	60.80	56.45	65.08
	Maximal lateral sway CoP O	9.83	62.40	46.77	77.78
	Sway Path C	485.44	67.20	62.90	71.43
	Anterior-Posterior Sway Path C	415.76	63.20	53.23	73.02
	Medial-Lateral Sway Path C	168.43	68.80	61.29	76.19
ROC	Sway Path O (AUC = 0.855)	365.00	76.00	80.65	71.43
	Anterior-Posterior Sway Path O (AUC = 0.835)	268.50	75.20	82.26	68.25
	Medial-Lateral Sway Path O (AUC = 0.842)	148.00	76.00	80.65	71.43
	Mean Amplitude O (AUC = 0.646)	3.25	58.40	83.87	33.33
	Anterior-Posterior Mean Amplitude (AUC = 0.697)	2.55	63.20	82.26	44.44
	Medial-Lateral Mean Amplitude (AUC = 0.617)	1.45	53.60	80.65	26.98
	Maxima anteroposterior sway CoP O (AUC = 0.714)	9.80	63.20	80.65	46.03
	Maximal lateral sway CoP O (AUC = 0.631)	5.45	53.60	80.65	26.98
	Sway Path C (AUC = 0.744)	353.00	66.40	80.65	52.38
	Anterior-Posterior Sway Path C (AUC = 0.69)	244.00	57.60	80.65	34.92
	Medial-Lateral Sway Path C (AUC = 0.76)	128.50	62.40	82.26	42.86
Discriminant Function	Formula	2.80	77.60	61.29	93.65

Note: O—open eyes; C-closed eyes; CoP—plantar center of pressure; AUC—Area Under Curve. Discriminant Function Formula: 0.0076 · Sway Path O + 0.0009 · Anterior-Posterior Sway Path O—0.0103 · Medial-Lateral Sway Path O—0.3857 · Mean Amplitude O + 0.2153 · Anterior-Posterior Mean Amplitude + 0.075 · Medial-Lateral Mean Amplitude + 0.0205 · Maxima anteroposterior sway CoP O + 0.0495 · Maximal lateral sway CoP O + 0.012 · Sway Path C—0.0146 · Anterior-Posterior Sway Path C + 0.0066 · Medial-Lateral Sway Path C.

## Data Availability

All data used in this study were stored at: https://repozytorium.ur.edu.pl/handle/item/5877 (accessed on 1 January 2021).

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
