# Peer review of "The Use of Static Posturography Cut-Off Scores to Identify the Risk of Falling in Older Adults"

_ijerph, 2022, doi:10.3390/ijerph19116480_

Round 1
Reviewer 1 Report
The authors have conducted a study to identify the cut-off scores of standing posturography measures which can be used to predict the risk of falls in older adults.
Overall, the study holds some significance in the field and has potential. However, the manuscript is poorly written, particularly the introduction and discussion and there are major issues that must be addressed.
The introduction is poorly written and lacks flow and must be rewritten, there are several grammatical errors and some sentence presents an idea that is different from the previous one. These major errors must be fixed all over the manuscript. Describe posturography in more detail as to how it works and why this would be an important tool and the rationale behind its ability to differentiate "fallers" and "nonfallers."
Methods: The approval number from the ethics committee must be provided. This must be at the beginning of the methods section.
Please write in a paragraph format rather than point by point.
Results: Please invert table 1 so that each group is in each column.
Discussion: The discussion is limited and poorly written without comparing and contrasting with previous literature. Currently, simply it is a reiteration of results.
The authors first said that the eyes closed test in men does not work on several postural meter parameters and then in the next sentence, they claim it can work in men and women. Such arguments/claims must be backed up by data from the study/previous literature.
For EO are three decimals necessary?
Author Response
Dear Reviewer,
I resubmit to you a second version of manuscript entitled “The use of static posturography cut-off scores to identify the risk of falling in older adults”. We are grateful to the editors and reviewers for your time and comments on our manuscript. Thank you for giving me the opportunity to revise again and resubmit this manuscript.
According to the comments raised by the reviewers, we modified manuscript.
We are now resubmitting the revised manuscript and also the point-by-point response to the comments. All changes are highlighted as red text in the manuscript. The manuscript has been proofread by a native English speaker; any minor edits in language are not highlighted in the text.
We hope you will be pleased with the changes, and support the publication of our revised manuscript.
With kind regards,
Agnieszka Wiśniowska-Szurlej
Response to Reviewer 1 Comments
The authors have conducted a study to identify the cut-off scores of standing posturography measures which can be used to predict the risk of falls in older adults.
Overall, the study holds some significance in the field and has potential. However, the manuscript is poorly written, particularly the introduction and discussion and there are major issues that must be addressed.
Point 1: The introduction is poorly written and lacks flow and must be rewritten, there are several grammatical errors and some sentence presents an idea that is different from the previous one. These major errors must be fixed all over the manuscript. Describe posturography in more detail as to how it works and why this would be an important tool and the rationale behind its ability to differentiate "fallers" and "nonfallers."
Response 1: Thank you for your valuable remark. The introduction has been supplemented as suggested as follows:
„ Falling is the most common accident that occurs in daily living and the second leading cause of unintentional injury death in the world [1]. Approximately 28-35% of people aged 65 years and older fall every year, and this result increases to 32-42% for people over 70 years of age [2]. In older people, falls can lead to severe injuries such as hip fracture or head trauma [3]. The length of hospitalization in older people is nine days longer for fall-related reasons than for other reasons. Moreover, older adults hospitalized due to fall-related injuries are more often moved into long-term care facilities [4]. These transfers result in a loss of independence in everyday activities for the individual and increased healthcare system costs.
Although falls can occur at any age, their frequency increases with age. Falls that are experienced by the older people result from involutional changes (physical, perceptual and cognitive) together with the inappropriate environment regarding its safety [5]. There are many validated tools to predict the risk of falling in the elderly. According to the systematic review of Seong-Hi Park the Berg Balance Scale (BBS) and Timed Up and Go (TUG) test were generally used for the community-dwelling older people as fall risk assessment tools [6].To identify older people at increased risk of future falls, studies have most commonly used a history of falls and the assessment of impairments in gait, mobility and balance. Although the systematic assessment of the risk of falls in the elderly is recommended in geriatric medicine, the tools currently are not tools allowing for a reliable and objective measurement result [7].
Research indicates that postural instability is best studied with objective posturographic measurements, the advantage of which over regular clinical evaluation include reducing the variability of test scores and avoiding subjective scoring systems [8]. Posturography provides information about specific balance control mechanisms and thus constitutes a clinically useful tool to identify fall risk. Posturographic tests use force plates to measure the movement of the foot center of pressure (CoP), while the subject stands in a relaxed position on the platform with his eyes fixed at one point [9]. The difficulty of taking the measurement can be enlarged by eliminating the visual control. The eyes closed increase the postural sway in people of all ages compared with eyes open, however the differences are most distinguishable in older adults. Maintaining the correct balance of body posture contributes to the prevention of falls and affects the longer functional independence of the older people [10].
The studies with the use of posturography have shown that it is an objective and quantitative measure of balance deficits and they have confirmed its important clinical application in the assessment of the risk of falling in older people [11, 12]. Some sway characteristics of stance, especially in the mediolateral direction, significantly differ between nonfallers and fallers and may therefore be good indicators of individuals at increased risk of future falls. To identify the differences between falling and nonfalling older people, studies have assessed anterior-posterior (AP) plantar centre of pressure (CoP) movements and medial-lateral (ML) sway amplitude with eyes open and closed [13,14]. Additionally, Pizzigalli et al. indicated that path length CoP, CoP velocity and sway in the AP and ML directions are the variables that distinguish older adult fallers from nonfallers [15]. A larger sway when the eyes are closed is characteristic of people who have fallen repeatedly [16]. Studies have also shown differences in automated postural responses between older men and women [17]. According to Howcroft et al, a posturography examination in the standing position in older adults is a useful screening tool for older people who are at risk of falling [18].
The authors emphasize that objective measures of postural sway independently predict incident falls in older women and men [19]. Despite the widespread use of stabilometric measurements, there is currently no consensus about what values of features should be used for diagnostics [20]. Therefore, additional studies are needed to determine the postural stability values important for predicting falls in a clinical setting. Measures of postural sway predict falls in the elderly. It is important to identify people most at risk of falling in order to implement the necessary precautions and prevent (or at least slow down) age-related causes of falls by means of training and rehabilitation [21].
To the best of our knowledge, this is the first study in Poland presenting eyes open and eyes closed standing posturography with older adults to determine cut-off scores for risk of falling. The aims of the study were to assess postural stability with the eyes open and closed in older women and men and to establish the cut-off scores that can be used to identify people with an increased risk of falling.“
Point 2: Methods: The approval number from the ethics committee must be provided. This must be at the beginning of the methods section.
Response 2: The number of the Bioethics Committee has been added at the beginning of the methods section:
„ This observational study was carried out among a randomly selected population of older people. The study was approved by the Bioethics Committee of the University of Rzeszow (Resolution No. 4/3/2017). In accordance with Declaration of Helsinki, the participants were provided with information about the aim and the course of the study, and expressed their written and informed consent to participate.
This study was performed and reported according to the Strengthening the Reporting of Observational Studies in Epidemiology Criteria (STROBE) [22].”
Point 3: Please write in a paragraph format rather than point by point.
Response 3: Thank you for your comment. The format of individual paragraphs has been prepared in accordance with the journal guidelines, therefore the point system has remained unchanged. Please accept it as it stands.
Point 4: Results: Please invert table 1 so that each group is in each column.
Response 4: Thank you for your suggestion. Table 1 has been corrected.
Point 5: Discussion: The discussion is limited and poorly written without comparing and contrasting with previous literature. Currently, simply it is a reiteration of results.
The authors first said that the eyes closed test in men does not work on several postural meter parameters and then in the next sentence, they claim it can work in men and women. Such arguments/claims must be backed up by data from the study/previous literature.
Response 5: Thank you for your remark. The discussion has been corrected as follows:
“Center of pressure trajectory parameters need to be identified to classify and predict people at risk of falling and to conduct objective and long-term monitoring of the most vulnerable elderly individuals. Recent advancements in the collection and storage of data enables posturographic data to be used to determine the magnitude of improvement in an individual’s ability to maintain balance in the process of rehabilitation and his or her status.
The results of this study show that posturographic data can distinguish people with a high risk of falling from people without risk of falling. The largest differences were related to the parameters of sway path, anterior-posterior sway path and medial-lateral sway path. Similar results were obtained by Merlo et al. The authors showed that the mean positions of the COP in the AP and ML directions were associated with fall history in older adults [30]. A systematic review by Quijoux et al. showed that prospective studies previously demonstrated that differences in the mean velocities in the AP and ML directions are significantly different between fallers and nonfallers. The ML features were less consistent and discriminative than were the AP features in the present analysis [31].
Among the examined women, all analysed posturographic parameters, both with eyes open and eyes closed, differed statistically significantly between HRF and WRF. In the men, a relationship was demonstrated between posturographic parameters with eyes open and the risk of falling. However, with eyes closed, no differences were found in the following parameters: mean amplitude, anterior-posterior mean amplitude, medi-al-lateral mean amplitude, maximal antero-posterior sway COP and maximal lateral sway COP. A prospective study by Cella et al. showed that stabilometric assessments can more accurately identify older people at high risk of falls, thus enabling more personalized fall prevention interventions [32].
Objective measures of postural sway allow for predicting risk of falling, although there are indeed no stability limits that can identify those who are most vulnerable in this regard. The results of the prospective observational study from the Healthy Aging Initiative cohort indicated that the parameters of postural sway would allow for an accurate fall risk prediction, whereas the outcomes of further work should set normative values in order to identify the most vulnerable people [33].
The research showed the highest sensitivity, specificity and accuracy for sway path. The area under the curve was the highest for a sway path of 0.89. The clinical cut-off score for sway path with EO was 350.63, and the area under the ROC was 368.50. According to Johansson et al., there was a 75% increase in fall risk in older adults with CoP sway lengths ≥400 mm in the test with open eyes [8], while Prosperini et al. indicated that the risk of falling may grow 8% to each 10-millimeter increase in the CoP pathway with eyes open [34].
In the group of women, the clinical cut-off score for sway path EO was 349.73 (ROC - 364.00), while for the men, the clinical cut-off score for sway path with EO was 358.06 (ROC – 365.00). This result means that people with the abovementioned results have a high risk of falling. Our study provides a basis for identifying, monitoring and managing interventions for elderly people at risk of falling.
Additionally, high values were shown for the anterior-posterior sway path. The clinical cut-off score for sway path with EO was 272.64, and the area under the ROC curve was 272.50. In the group of women, the clinical cut-off score for sway path with EO was 264.39 (ROC – 272.50), while for the men, the clinical cut-off score for sway path with EO was 286.78 (ROC – 268.50). As we observed higher values of sensitivity, specificity and ac-curacy for the clinical cut-off scores, the clinical cut-off score method demonstrated greater predictive value than did the ROC method.
This study has limitations. Due to the observational nature of the study, the idea that balance impairment precedes falls cannot be confirmed. The validity of the estimated cut-off points should be verified in a longitudinal study. Additionally, the limitation may be a convenient sample of relatively healthy older adults. Another limitation of the study is the lack of monitoring of actual falls as well as of existing chronic diseases and medi-cations.
To the best of our knowledge, this study helps fill an existing gap for rehabilitation experts by providing cut-off scores of postural balance measures for screening community-dwelling older adults for a high risk of falling. Identifying the most vulnerable people to a fall risk will help to implement proper training at an early stage. According to the systematic review of Sherrington et al. performing exercises reduces the frequency of falls by 23% [35]. Various forms of training, including multi-component exercise training [36], Tai Chi [37] or functional training with Total Resistance Exercises (TRX) lead to positive changes in the static and dynamic balance of older adults [38].
Point 6: For EO are three decimals necessary?
Response 6: Thank you for your remark. Changes have been made to the records of the results for EO with two decimals .

Reviewer 2 Report
This manuscript describes a study that aimed to identify the cut-off scores of standing posturography measures that can be used to predict the risk of falling in older adults.
Line 48: Change "(TUG0)" to "(TUG)"
Lines 48-49: Change "...and quite diverse 24 fall..." to "...and 24 quite diverse fall..."
Line 56: Change this to "Therefore it has been suggested that stabilometric tests may solve this problem."
Lines 70 and 72: Centre of Pressure is shortened to "CoP" here then "COP" throughout the rest of the manuscript. Best to stick with one or the other. My preferance would be "CoP" as I have seen that most commonly used.
Line 90: Perhaps replacing "identify" with "establish"
Line 108: Change "up-and-go test" to "timed up-and-go test"
Line 161 or 328: Please include the ethics approval number.
The Methods and Result sections are presented clearly.
Line 264: Change "COP" to "Centre of Pressure" to begin the section.
This study addressed risk of falling in older volunteers. I think a major flaw in the study is that all assessments were of static measures of CoP, eyes open or closed. Falls are more likely to occur while walking so I see no convincing justification for this methods of predicting risk of falling. There should be, at least, a correlation analysis between TUG and CoP measures.
The main problem I have with this study is that the measures are assessments of static balance, when falls are associated, more often, with tripping or stumbling while walking. TUG was assessed only for the purpose of separating participants into HRF and WRF although the authors do not make the connection between the TUG, a test of speed, and static balance and how this relate to fall risk. The authors, therefore, have not made a case for static measures having predictive power of falls risk. It may be more useful to split the groups based on history of falls rather than the TUG score.
Author Response
Dear Reviewer,
I resubmit to you a second version of manuscript entitled “The use of static posturography cut-off scores to identify the risk of falling in older adults”. We are grateful to the editors and reviewers for your time and comments on our manuscript. Thank you for giving me the opportunity to revise again and resubmit this manuscript.
According to the comments raised by the reviewers, we modified manuscript.
We are now resubmitting the revised manuscript and also the point-by-point response to the comments. All changes are highlighted as red text in the manuscript. The manuscript has been proofread by a native English speaker; any minor edits in language are not highlighted in the text.
We hope you will be pleased with the changes, and support the publication of our revised manuscript.
With kind regards,
Agnieszka Wiśniowska-Szurlej
Response to Reviewer 2 Comments
This manuscript describes a study that aimed to identify the cut-off scores of standing posturography measures that can be used to predict the risk of falling in older adults.
Point 1: Line 48: Change "(TUG0)" to "(TUG)"
Lines 48-49: Change "...and quite diverse 24 fall..." to "...and 24 quite diverse fall..."
Line 56: Change this to "Therefore it has been suggested that stabilometric tests may solve this problem."
Lines 70 and 72: Centre of Pressure is shortened to "CoP" here then "COP" throughout the rest of the manuscript. Best to stick with one or the other. My preferance would be "CoP" as I have seen that most commonly used.
Line 90: Perhaps replacing "identify" with "establish"
Line 108: Change "up-and-go test" to "timed up-and-go test"
Line 161 or 328: Please include the ethics approval number.
The Methods and Result sections are presented clearly.
Line 264: Change "COP" to "Centre of Pressure" to begin the section.
Response 1: Thank you for your remarks. These manuscript changes have been made.
Point 2: This study addressed risk of falling in older volunteers. I think a major flaw in the study is that all assessments were of static measures of CoP, eyes open or closed. Falls are more likely to occur while walking so I see no convincing justification for this methods of predicting risk of falling. There should be, at least, a correlation analysis between TUG and CoP measures.
The main problem I have with this study is that the measures are assessments of static balance, when falls are associated, more often, with tripping or stumbling while walking. TUG was assessed only for the purpose of separating participants into HRF and WRF although the authors do not make the connection between the TUG, a test of speed, and static balance and how this relate to fall risk. The authors, therefore, have not made a case for static measures having predictive power of falls risk. It may be more useful to split the groups based on history of falls rather than the TUG score.
Response 2: Thank you for your valuable remark. It is obvious that the balance of the body can be measured dynamically and statically. While planning the study, the research team reviewed the literature, according to which static posturography was used to identify people at a risk of falling [1]. It should be noted that the definitions of fall risk in the studies were not identical and were based, among others, on the following: distinguishing criteria of fall risk by Tinetii [2], Kellogg [3], CERAD [4], while others did not specify a specific fall risk criterion [5]. According to the researchers, the measurement of sway velocity shows excellent accuracy and is an appropriate tool for predicting future falls in older adults [6]. Earlier studies have also revealed that postural stability deficits during quiet standing are associated with the deterioration of postural control during complex motor tasks [7]. Therefore, we have decided to perform the research to determine precisely the cut-off points of postural stability, enabling the identification of older people with an increased risk of falling based on the TUG test results so as to implement preventive actions as early as possible in medical care facilities. In further research, we are going to complete the measurement of static balance with the analysis of balance under dynamic conditions.
- Piirtola M, Era P. Force platform measurements as predictors of falls among older people: A review. Gerontology. 2006;52: 1–16.
- Kurz I, Oddsson L, Melzer I. Characteristics of balance control in older persons who fall with injury—a prospective study. J Electromyogr Kinesiol. 2013;23: 814–819.
- Stel VS, Smit JH, Pluijm SMF, Lips P. Balance and mobility performance as treatable risk factors for recurrent falling in older persons. J Clin Epidemiol. 2003;56: 659–668.
- Merlo A, Zemp D, Zanda E, Rocchi S, Meroni F, Tettamanti M, et al. Postural stability and history of falls in cognitively able older adults: The Canton Ticino study. Gait Posture. 2012;36: 662–666.
- Maranesi E, Merlo A, Fioretti S, Zemp DD, Campanini I, Quadri P. A statistical approach to discriminate between non-fallers, rare fallers, and frequent fallers in older adults based on posturographic data. Clin Biomech. 2016;32: 8–13.
- Kwok BC, Clark RA, Pua YH. Novel use of the Wii Balance Board to prospectively predict falls in community-dwelling older adults. Clin Biomech (Bristol, Avon). 2015;30(5):481-4.
- Michalska J, Kamieniar A, Sobota G. et al.Age-related changes in postural control in older women: transitional tasks in step initiation. BMC Geriatr. 2021;21:17.

Reviewer 3 Report
This manuscript describes a quantitative observational investigation of the relationship between the TUG score and the posturography measures in older persons. The authors comprehensively explored cut-off scores of the posturography measures that separate low and high fall risk groups. Although the quantitative analysis with the large sample number in this study will provide insightful perspectives for fall-risk assessments using stabilometric tests, there are some issues that should be revised before publication.
- Please mention the use of TUG for target grouping in the abstract.
- There is no description on the reason the three classification methods were selected for the comparison. Please clarify it.
- Please reconsider the significant figures of the values shown as results. Some of them are six and seem to be too large.
- In L298-299, the AUC value seems to be wrong.
Author Response
Dear Reviewer,
I resubmit to you a second version of manuscript entitled “The use of static posturography cut-off scores to identify the risk of falling in older adults”. We are grateful to the editors and reviewers for your time and comments on our manuscript. Thank you for giving me the opportunity to revise again and resubmit this manuscript.
According to the comments raised by the reviewers, we modified manuscript.
We are now resubmitting the revised manuscript and also the point-by-point response to the comments. All changes are highlighted as red text in the manuscript. The manuscript has been proofread by a native English speaker; any minor edits in language are not highlighted in the text.
We hope you will be pleased with the changes, and support the publication of our revised manuscript.
With kind regards,
Agnieszka Wiśniowska-Szurlej
Response to Reviewer 3 Comments
This manuscript describes a quantitative observational investigation of the relationship between the TUG score and the posturography measures in older persons. The authors comprehensively explored cut-off scores of the posturography measures that separate low and high fall risk groups. Although the quantitative analysis with the large sample number in this study will provide insightful perspectives for fall-risk assessments using stabilometric tests, there are some issues that should be revised before publication.
Point 1: Please mention the use of TUG for target grouping in the abstract.
Response 1: Thank you for your remark. The change has been introduced in the summary.
Point 2: There is no description on the reason the three classification methods were selected for the comparison. Please clarify it.
Response 2: Thank you for your comment. We have selected three different factors for the analysis in order to increase the likelihood of identifying the most important and the best cut-off point for the results of static posturography. The Clinical Cut of Score is considered to be the authoritative cut-off score for clinically significant differences in the study groups [1]. The method of determining ROC curves allows researchers to identify cut-off points of the desired sensitivity and specificity [2]. In contrast, discriminant analysis enables a linear classification between the two study groups (with and without risk of falling) and is similar to logistic regression [3]. Linear discriminant function was used to predict for which values of the variables the studied groups differed the most.
- Thomas JC, Truax P. Assessment and analysis of clinically significant change In: McKay D, editor. Handbook of research methods on abnormal and clinical psychology. California: SAGE Publications, Inc; 2008;319–32
2. Hajian-Tilaki K. Receiver Operating Characteristic (ROC) Curve Analysis for Medical Diagnostic Test Evaluation. Caspian J Intern Med. 2013;4(2):627-635.
3. Pohar M, Blas M, Turk S. Comparison of logistic regression and linear discriminant analysis: A simulation study. Metodoloski zvezki. 2004;1: 143–161
Point 3: Please reconsider the significant figures of the values shown as results. Some of them are six and seem to be too large.
Response 3: Thank you for your remark. The data in the text has been modified to include numerical values up to two decimal places.
Point 4: In L298-299, the AUC value seems to be wrong.
Response 4: Thank you for your remark, the indicated text fragment has been checked.

Reviewer 4 Report
Congratulations for this manuscript, i have add some comments to improve this manuscript.
INTRODUCTION
It's very correct, but you can add information about exercise program and risk of falling and dynamic balance, etc.. TRX and risk of falling in elderly and other programs.
METHODS
line 107 " A group..." should the letter A.
Have you adjusted the statistical analysis by basic confounders? (age, sex, level education and IMC) You should add this information into the manuscript.
RESULTS
You should add one figure in the "results" section to understand all the results, however the results are very good.
DISCUSSION
Its correct in all parts.
Author Response
Dear Reviewer,
I resubmit to you a second version of manuscript entitled “The use of static posturography cut-off scores to identify the risk of falling in older adults”. We are grateful to the editors and reviewers for your time and comments on our manuscript. Thank you for giving me the opportunity to revise again and resubmit this manuscript.
According to the comments raised by the reviewers, we modified manuscript.
We are now resubmitting the revised manuscript and also the point-by-point response to the comments. All changes are highlighted as red text in the manuscript. The manuscript has been proofread by a native English speaker; any minor edits in language are not highlighted in the text.
We hope you will be pleased with the changes, and support the publication of our revised manuscript.
With kind regards,
Agnieszka Wiśniowska-Szurlej
Response to Reviewer 4 Comments
Congratulations for this manuscript, i have add some comments to improve this manuscript.
Thank you for taking the time to review our manuscript.
Point 1: INTRODUCTION
It's very correct, but you can add information about exercise program and risk of falling and dynamic balance, etc.. TRX and risk of falling in elderly and other programs.
Response 1: Thank you for your remark. Due to the numerous changes made in the introduction in line with the reviewers' comments, we have decided to enrich the discussion with several studies related to exercise programs that reduce the risk of falling.
Identifying the most vulnerable people to fall risk will help to implement proper training at an early stage. According to the systematic review of Sherrington et al. performing exercises reduces the frequency of falls by 23% [35]. Various forms of training, including multicomponent exercise training [36], Tai Chi [37] or functional training with Total Resistance Exercises (TRX) lead to positive changes in the static and dynamic balance of older adults [38].
Point 2: METHODS
line 107 " A group..." should the letter A.
Response 2: Thank you for your remark. The change has been made.
Point 3: Have you adjusted the statistical analysis by basic confounders? (age, sex, level education and IMC) You should add this information into the manuscript.
Response 3: Thank you for your remark. In collaboration with a biostatistician, a statistical analysis was performed that did not take into account confounding factors, whereas the cut-off points for postural stabilometry were calculated separately for women and men.
Point 4: RESULTS
You should add one figure in the "results" section to understand all the results, however the results are very good.
Response 4: Thank you for your remark. After a careful analysis of the obtained research results, we decided that it was not possible to present such extensive data using a graph. Please accept the current form of data presentation.
DISCUSSION
Its correct in all parts.

Round 2
Reviewer 1 Report
The authors have submitted a revised version of their manuscript. The authors have tried to address my concerns. However, I believe that the discussion can be improved. The authors have done a great job of comparing and contrasting previous studies with their current findings, however, they only discussed two studies. More studies must be discussed to compare findings from their study. If the authors believe that there are no studies that are relevant to discuss, this should be mentioned and addressed as a strength of the paper. Discussion must indicate this point.
Author Response
Dear Reviewer,
I resubmit to you a third version of manuscript entitled “The use of static posturography cut-off scores to identify the risk of falling in older adults”. We take this opportunity to express our gratitude to the reviewers for their constructive and useful remarks. Their comments allowed us to identify areas in our manuscript that needed modification and clarification.
We hope you will be pleased with the changes, and support the publication of our revised manuscript.
With kind regards,
Agnieszka Wiśniowska-Szurlej
Response to Reviewer 1 Comments
Point 1: The authors have submitted a revised version of their manuscript. The authors have tried to address my concerns. However, I believe that the discussion can be improved. The authors have done a great job of comparing and contrasting previous studies with their current findings, however, they only discussed two studies. More studies must be discussed to compare findings from their study. If the authors believe that there are no studies that are relevant to discuss, this should be mentioned and addressed as a strength of the paper. Discussion must indicate this point.
Response 1: Thank you for your valuable comments and the opportunity to improve our manuscript. After an extensive review of publications analyzing the topic of our study, we have concluded that this is the first study of this type. Therefore, as suggested by the Reviewer, we have added the following paragraph to the discussion:
„The strength of this study is that, to the best of our knowledge, it is the first research assessing postural stability with the eyes open and closed in older women and men and establishing the cut-off scores that can be used to identify people with an increased risk of falling. Although several studies had previously agreed that older fallers exhibit higher COP displacements than older non-fallers, the ranges distinguishing people with an increased risk of falling still remained undefined. Therefore, this study helps fill an existing gap for rehabilitation experts by providing cut-off scores of postural balance measures for screening community-dwelling older adults for a high risk of falling. Identifying the most vulnerable people to a fall risk will help to implement proper training at an early stage.”

Reviewer 2 Report
I find the manuscript vastly improved from the previous version. My main concern was that the static tests were not strong predictors of dynamic balance, however the authors have added descriptions that make the connection clear.
I have no further suggestions for this manuscript.
Author Response
Dear Reviewer,
I resubmit to you a third version of manuscript entitled “The use of static posturography cut-off scores to identify the risk of falling in older adults”. We are grateful to the editors and reviewers for your time and comments on our manuscript.
With kind regards,
Agnieszka Wiśniowska-Szurlej
Response to Reviewer 2 Comments
Point 1: I find the manuscript vastly improved from the previous version. My main concern was that the static tests were not strong predictors of dynamic balance, however the authors have added descriptions that make the connection clear.
I have no further suggestions for this manuscript.
Response 1: Thank you very much for accepting the changes that have been made to the manuscript.

Reviewer 3 Report
The authors have solved the issues raised by the reviewer.
Author Response
Dear Reviewer,
I resubmit to you a third version of manuscript entitled “The use of static posturography cut-off scores to identify the risk of falling in older adults”. We are grateful to the editors and reviewers for your time and comments on our manuscript.
With kind regards,
Agnieszka Wiśniowska-Szurlej
Response to Reviewer 3 Comments
Point 1: The authors have solved the issues raised by the reviewer.
Response 1: Thank you very much for accepting the changes that have been made to the manuscript.
